# Expression of Nectin-4 and PD-L1 in Upper Tract Urothelial Carcinoma

**DOI:** 10.3390/ijms21155390

**Published:** 2020-07-29

**Authors:** Eisuke Tomiyama, Kazutoshi Fujita, Maria Del Carmen Rodriguez Pena, Diana Taheri, Eri Banno, Taigo Kato, Koji Hatano, Atsunari Kawashima, Takeshi Ujike, Motohide Uemura, Tetsuya Takao, Seiji Yamaguchi, Hiroaki Fushimi, Kazuhiro Yoshimura, Hirotsugu Uemura, George J. Netto, Norio Nonomura

**Affiliations:** 1Department of Urology, Osaka University Graduate School of Medicine, Osaka 565-0871, Japan; tomiyama@uro.med.osaka-u.ac.jp (E.T.); kato@uro.med.osaka-u.ac.jp (T.K.); hatano@uro.med.osaka-u.ac.jp (K.H.); kawashima@uro.med.osaka-u.ac.jp (A.K.); ujike@uro.med.osaka-u.ac.jp (T.U.); uemura@uro.med.osaka-u.ac.jp (M.U.); nono@uro.med.osaka-u.ac.jp (N.N.); 2Department of Urology, Kindai University Faculty of Medicine, 377-2, Ohno-Higashi, Sayama, Osaka 589-8511, Japan; ebanno@med.kindai.ac.jp (E.B.); yoshimur@med.kindai.ac.jp (K.Y.); huemura@med.kindai.ac.jp (H.U.); 3Department of Pathology, Johns Hopkins University, Baltimore, MD 21287, USA; 4Department of Pathology, University of Alabama at Birmingham, Birmingham, AL 35233-7331, USA; mdcrodriguez@uabmc.edu (M.D.C.R.P.); gnetto@uabmc.edu (G.J.N.); 5Department of Pathology, Kidney Diseases Research Center, Isfahan University of Medical Sciences, Isfahan 8174673461, Iran; diana1380@yahoo.com; 6Department of Urological Immuno-Oncology, Osaka University Graduate School of Medicine, Osaka 565-0871, Japan; 7Department of Urology, Osaka General Medical Center, Osaka 558 8558, Japan; takao@gh.opho.jp (T.T.); syamaguchi@gh.opho.jp (S.Y.); 8Department of Pathology, Osaka General Medical Center, Osaka 558 8558, Japan; hiroaki-fushimi@gh.opho.jp

**Keywords:** Nectin-4, Programmed Death Ligand 1, upper tract urothelial carcinoma, enfortumab vedotin

## Abstract

Enfortumab vedotin is a novel antibody–drug conjugate targeting Nectin-4, which is highly expressed in urothelial carcinoma. However, the expression status of Nectin-4 in upper tract urothelial carcinoma (UTUC) remains unclear. The relationship between Nectin-4 and Programmed Death Ligand 1 (PD-L1) in UTUC is also ambiguous. We performed immunohistochemical analysis of 99 UTUC tissue microarray to assess the expression of Nectin-4 and PD-L1 in UTUC. Nectin-4-positivity was detected in 65 (65.7%) samples, and PD-L1 was detected in 24 (24.2%) samples. There was no correlation between the expression of Nectin-4 and PD-L1. Patients with strong Nectin-4-expressing tumors had a significantly higher risk of progression (*p* = 0.031) and cancer-specific mortality (*p* = 0.036). Strong Nectin-4 expression was also an independent predictor of disease progression in the high-risk group (pT3 ≤ or presence of lymphovascular invasion or lymph node metastasis) (Hazard ratio, 3.32 [95% confidence interval, 1.20–7.98; *p* = 0.027]). In conclusion, we demonstrated that Nectin-4 expression rate in UTUC was 65.7% and independent of PD-L1 expression. Strong Nectin-4 expression was associated with worse progression-free survival in high-risk UTUC. These findings suggested that enfortumab vedotin may be effective in a broad range of patients with UTUC, regardless of PD-L1 expression.

## 1. Introduction

Upper tract urothelial carcinoma (UTUC) is relatively rare, accounting for 5%–10% of all urothelial carcinomas [1,2]. The natural history of UTUC differs from that of urinary bladder urothelial carcinoma (UBC); patients with UTUC often present with invasion or metastases at diagnosis and have a poorer prognosis than those with UBC [1]. While the morphology of UTUC is similar to that of UBC, phenotypical and genetic differences between the two have been reported [2]. Increasing evidence suggests that UTUC could be considered a distinct disease entity from UBC and highlights the importance of developing optimal management strategies for this disease. Locally advanced or metastatic urothelial carcinoma, including UTUC, is an incurable disease with poor survival. Currently, the first-line therapy for UTUC includes platinum-based chemotherapy, while anti-Programmed Death 1 (PD-1) or Programmed Death Ligand 1 (PD-L1) inhibitor is the second-line therapy for these patients [3]. Treatment options are limited in case of patients who show tumor progression even after platinum-based chemotherapy and/or PD-1/PD-L1 inhibitor therapy.

Nectins are immunoglobulin-like transmembrane proteins that are found in the adherens junctions of cells and mediate Ca^2+^-independent cell–cell adhesion via both homophilic and heterophilic trans interactions [3]. Recent phase I and II trials demonstrated the efficacy of enfortumab vedotin (EV), which targets Nectin-4 in urothelial carcinoma patients who exhibit tumor progression after platinum chemotherapy and/or immune checkpoint inhibitors [4,5]. EV is an antibody-drug conjugate, comprised of the human anti-Nectin-4 antibody and monomethyl auristatin E (MMAE), an anti-microtubule agent [4]. Nectin-4 is highly expressed in urothelial, breast, lung, and pancreatic carcinomas, and EV binds to cells that express Nectin-4 with high affinity, initiating the internalization and release of MMAE in the target cells [3]. Furthermore, patients who did not respond to PD-1/PD-L1 inhibitor responded to EV in these trials. Thus, EV could be a promising treatment option for patients with locally advanced or metastatic urothelial carcinoma that progresses after treatment with platinum-based chemotherapy and/or PD-1/PD-L1 inhibitor. However, Nectin-4 expression in UTUC has not been studied in detail, and to our knowledge, there are no reports on the association between the expression of Nectin-4 and PD-L1 in UTUC.

In the present study, we investigated the expression of Nectin-4 in UTUC using tissue microarray (TMA) specimens of patients with UTUC. We also studied the association between Nectin-4 protein expression and clinicopathological features or outcomes. Additionally, we also examined PD-L1 expression and its correlation with that of Nectin-4. Thus, our study could help identify patients with UTUC who may benefit from EV treatment.

## 2. Results

### 2.1. Patient Characteristics

The clinicopathological characteristics and outcomes of the 99 patients included in the study are summarized in Table 1.

Concomitant radical metastasectomy was performed for two patients who had peritoneal dissemination and distant lymph node metastases (mesentery lymph nodes). None of the tumors showed squamous or glandular differentiation or other histological types such as small cell carcinoma. Tumor progression was defined as the development of non-lower urinary tract lesions, including recurrence at the nephroureterectomy site and lymph node or visceral metastasis. None of the patients received neoadjuvant therapy before the collection of the tissues included in the TMA; however, 26 patients (26.3%) underwent 2 to 3 cycles of adjuvant chemotherapy with MVAC (methotrexate, vinblastine, adriamycin, and cisplatin).

### 2.2. Immunohistochemical Analysis of Nectin-4 and PD-L1 Expression in UTUC TMA

We conducted immunohistochemical analysis of Nectin-4 expression in 99 UTUC samples and corresponding non-neoplastic urothelium. Typical patterns of Nectin-4 immunohistochemical expression in UTUC TMA specimens are shown in Figure 1.

Nectin-4-expression was detected in 65 (65.7%) of 99 UTUC samples (1+, 31.3%; 2+, 24.2%; and 3+, 10.1%), which was significantly higher than that in non-neoplastic urothelium (26.9%; 1+, 20.5%; 2+, 5.1%; and 3+, 1.3%; *p* < 0.001). We also conducted immunohistochemical analysis of PD-L1 expression in the 99 UTUC samples. Immunohistochemical PD-L1 expression is depicted in Figure 2.

PD-L1 expression was positive in 24 (24.2%) of 99 patients with UTUC. Next, we analyzed the associations of Nectin-4 and PD-L1 expression with clinicopathological profiles of 99 patients with UTUC (Table 2).

There was no association between Nectin-4 expression status and patient sex, location, stage, grade, and presence of lymphovascular invasion or lymph node metastasis (0 vs. 1+/2+/3+, 0 vs. 3+). PD-L1 positivity was significantly higher in females (*p* = 0.034), whereas there was no association between PD-L1 positivity and other clinicopathological characteristics (Appendix A). Notably, Nectin-4 expression was also not significantly associated with PD-L1 expression (*p* =0.806), and 15 (15.2%) of the 99 cases were positive for both Nectin-4 and PD-L1 (Figure 3).

### 2.3. Association of Nectin-4 and PD-L1 Expression with Patient Prognosis

We then performed Kaplan–Meier analysis to assess the prognostic value of Nectin-4 expression in UTUC. We found no significant difference in tumor progression (Figure 4A) or cancer-specific mortality (Figure 4B) in patients with Nectin-4 positive tumors and those with Nectin-4 negative tumors. However, patients with strong Nectin-4 expression in tumors had a significantly higher risk of tumor progression (Log-rank test, *p* = 0.031; Figure 4C) or cancer-specific mortality (Log-rank test, *p* = 0.036; Figure 4D) than patients who did not show strong Nectin-4 expression in tumors (0/1+/2+ vs. 3+). A comparison between patients with strong Nectin-4 expressing tumors and those with Nectin-4 negative tumors for tumor progression or cancer-specific mortality (0 vs. 3+) is shown in Appendix A.

To determine whether strong Nectin-4 expression was an independent prognostic factor in patients with UTUC, we performed a multivariate analysis with the Cox model. In the entire cohort, strong Nectin-4 expression was not an independent predictor of cancer-specific mortality, but it tended to be associated with a higher risk of tumor progression (Table 3).

We defined high-risk group as patients with pathological stage higher than pT3 or presence of lymphovascular invasion or positive lymph nodes. In this group (*n* = 50), strong Nectin-4 expression was found to be an independent predictor of tumor progression (Table 4).


In contrast, the expression status of PD-L1 did not show a significant association with tumor progression (*p* = 0.254) or cancer-specific mortality (*p* = 0.381) (Appendix A).

## 3. Discussion

Given that the efficacy of PD-1/PD-L1 inhibitors was limited in the second-line setting (the objective response rates were only 15.0–21.1% [6,7,8,9]), a new therapeutic option is eagerly expected for UTUC. In December 2019, The Food and Drug Administration (FDA) approved EV for treatment of patients with locally advanced or metastatic urothelial carcinoma, who previously received PD-1/PD-L1 inhibitor and/or platinum-containing chemotherapy. As EV is an antibody-drug conjugate targeting Nectin-4, the expression of Nectin-4 in cancer cells is necessary for its cytotoxic effect. In this study, we immunohistochemically analyzed the expression of Nectin-4 in UTUC and found it to be expressed in about 65.7% of the total samples.

So far, Nectin-4 has been considered to be overexpressed in several types of cancer [3]. In urothelial carcinoma, Nectin-4 expression were detected in 434 (82.8%) of 524 bladder cancers (1+, 118 [22.5%]; 2+, 154 [29.4%]; and 3+,162 [30.9%]) [3], and another study reported that Nectin-4 expression level (median H-score) was 290(range:14–300) in 125 UCs including 44 UTUCs (35%) [4]. However, these Nectin-4 expression data of urothelial carcinoma were investigated predominantly in bladder cancer, and the expression status in UTUC remains unclear. Our findings indicated that UTUC might have a slightly lower rate of Nectin-4 positivity than that of UBC [3,4].

Overexpression of Nectin-4 is reportedly implicated in disease progression and poor prognosis in several types of cancer [10,11,12,13]. However, the prognostic significance of Nectin-4 expression has not been investigated in urothelial carcinoma, including UTUC, to date.

For the understanding of the function of Nectin-4 in cancer, Zhang et al. reported that Nectin-4 regulates Rac-1 activity by activating the phosphoinositide 3-kinase (PI3K)/protein kinase B (AKT) signaling pathway to mediate cell proliferation and migration in gallbladder carcinoma [13]. In the present study, strong Nectin-4 expression was shown to be associated with worse progression-free survival in high-risk UTUC group. Although the benefit of adjuvant cisplatin-based chemotherapy for patients with high-risk UTUC is still controversial [1,14,15], EV may be useful as an adjuvant therapy for patients with high-risk UTUC expressing high levels of Nectin-4.

PD-1/PD-L1 inhibitor and EV are FDA-approved treatments for locally advanced or metastatic urothelial carcinoma after platinum-containing chemotherapy. It would be clinically important to examine the relationship between the target proteins of each drug. Two phase III trials (IMvigor130 for atezolizumab and KEYNOTE-361 for pembrolizumab) comparing PD-L1 inhibitor with platinum-based chemotherapy in the first-line setting suggested that PD-L1 inhibitor was inferior to platinum-based chemotherapy when PD-L1 expression was low [16,17]. Therefore, effective treatment options for PD-L1-negative patients may be limited. FDA has limited the use of pembrolizumab or atezolizumab only for patients with PD-L1-expressing locally advanced or metastatic urothelial cancer who are not eligible for cisplatin-containing therapy [17]. As 76% (75 of 99) cases of UTUC in the present study were negative for PD-L1 expression, only 24% of patients with UTUC would be eligible for the pembrolizumab or atezolizumab treatment. However, we also demonstrated Nectin-4 expression was independent of PD-L1 expression. Therefore, EV may be effective in patients with PD-L1-negative UTUC. Indeed, the EV-201 study (phase II trial) also showed a response of EV to urothelial carcinomas, irrespective of the previous response to anti–PD-1/L1 therapy [4]. These data demonstrate the ability of EV to elicit responses across a broad range of patients with UTUC, irrespective of the PD-L1 expression level.

The present study has several limitations. First, the expression status of Nectin-4 in primary and metastatic foci may be different. Besides, UTUC specimens in the present study, which were obtained by radical nephroureterectomy, included low-risk UTUC as well. The metastatic UTUC possessing a more aggressive phenotype may exhibit higher Nectin-4 expression than that in local UTUC. Therefore, the actual response rate of EV in metastatic UTUC needs to be confirmed in future clinical studies. The association of the therapeutic effect of EV with Nectin-4 expression in UTUC also needs to be studied. Second, the differences in immunohistochemical staining methods (e.g., primary antibodies) could affect the result of Nectin-4 expression analysis. Other factors, such as the aging of TMA, could also affect the rate of Netin4 positivity. All TMA specimens in the current study were from Japanese patients; this racial difference also might affect the expression. Third, immunohistochemical analysis is a semi-quantitative evaluation method, and may not have represented the actual expression of these markers.

In conclusion, we demonstrated that Nectin-4 expression rate in UTUC was 65.7% and independent of PD-L1 expression. Strong expression of Nectin-4 was associated with worse progression-free survival in high-risk UTUC. These findings suggested that EV, an antibody-drug conjugate targeting Nectin-4, would be effective in a broad range of patients with UTUC irrespective of the PD-L1 expression level.

## 4. Materials and Methods

### 4.1. Patients and Tissue Samples

The UTUC TMA was constructed with spotted triplicate urothelial tumor samples (from dominant tumors/invasive components if present) and paired normal-appearing urothelial tissues (from the renal pelvis and ureter) obtained from 99 patients with non-metastatic UTUC, who underwent radical nephroureterectomy performed at Osaka General Medical Center, Osaka between 1997 and 2011, as described previously [18,19,20,21,22]. Appropriate approval was obtained from the local institutional review board (Osaka General Medical Center Institutional Review Board, Protocol Number: 25-2014, 19 June 2013) before construction and use of the TMA, and written informed consent was obtained from all patients.

### 4.2. Immunohistochemistry

Immunohistochemical staining for Nectin-4 and PD-L1 was performed on the tissue sections (5-μm-thick) from the UTUC TMA using the EnVision system (DAKO, Glostrup, Denmark), according to the manufacturer’s instructions. The tissue sections were deparaffinized using xylene and a graded series of ethanol. For Nectin-4 antigen retrieval, the sections were treated with Tris-ethylenediaminetetraacetic acid (EDTA) buffer (pH 9.0) and steamed by placing them above boiling water for 20 min; PD-L1 antigen was retrieved by treatment with Tris-EDTA buffer (pH 9.0) at 120 °C for 10 min in a pressure cooker (Pascal; DAKO, Glostrup, Denmark). Endogenous peroxidase activity was blocked with 0.3% hydrogen peroxide for 5 min. Primary antibodies against Nectin-4 (1:3000; EPR15613-68; Abcam, Cambridge, UK) and PD-L1 (1:100; E1L3N; Cell Signalling, Danvers, MA, USA) were added and incubated overnight at 4 °C. Then, we used EnVision + System-HRP labeled polymer anti-rabbit (DAKO, Glostrup, Denmark) according to the manufacturer’s instructions. Sections were counterstained with hematoxylin, dehydrated in a graded series of ethanol, and cleared in xylene, and a coverslip was placed on them.

### 4.3. Scoring System

The intensity and extent of Nectin-4 expression were determined microscopically using the histochemical scoring system (H-score), which was defined as the product of the staining intensity (score, 0–3) and percentage of stained cells (0–100) at a given intensity. Specimens were then classified as negative (0; H-score, 0–14), weak (1+; H-score, 15–99), moderate (2+; H-score, 100–199), and strong (3+; H-score, 200–300).

The extent of membranous PD-L1 expression was also determined microscopically in each spot (0–100%). The average PD-L1 expression was calculated in each case and a 1% positivity cut-off was used.

### 4.4. Statistical Analyses

Statistical analyses were performed using JMP^®^ Pro 14.0.0 (SAS Institute Inc., Cary, NC, USA), and data were visualized using GraphPad Prism version 7.05 (GraphPad Software, San Diego, CA, USA). Fisher’s exact test was used to evaluate the association between categorized variables. The survival rates were determined using the Kaplan–Meier method, and the log-rank test was used for comparison. The Cox proportional hazards model was used to determine the statistical significance of prognostic indicators in univariate and multivariate settings. *p*-values < 0.05 were considered statistically significant, and <0.1 were considered to be statistically trending.

## Figures and Tables

**Figure 1 ijms-21-05390-f001:**
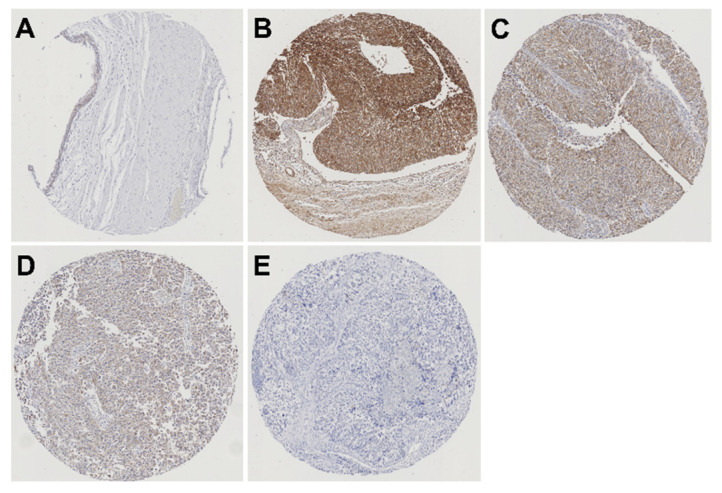
Typical patterns of immunohistochemical expression of Nectin-4 in UTUC tissue microarray specimens. (**A**) Normal urothelium with weak expression. (**B**) UTUC tissue with strong expression. (**C**) UTUC tissue with moderate expression. (**D**) UTUC tissue with weak expression. (**E**) UTUC tissue with negative expression.

**Figure 2 ijms-21-05390-f002:**
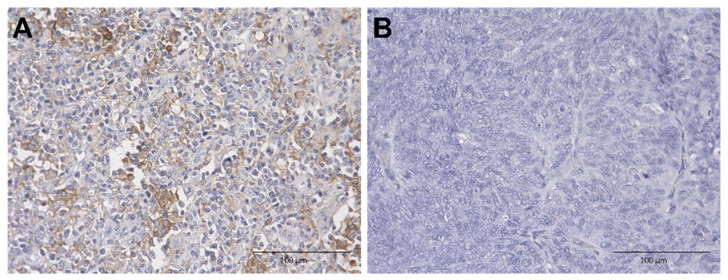
Membranous Programmed Death Ligand 1 (PD-L1) expression in upper tract urothelial carcinoma (UTUC) tissue microarray specimens. (**A**) UTUC tissue with moderate expression. (**B**) UTUC tissue with negative expression.

**Figure 3 ijms-21-05390-f003:**
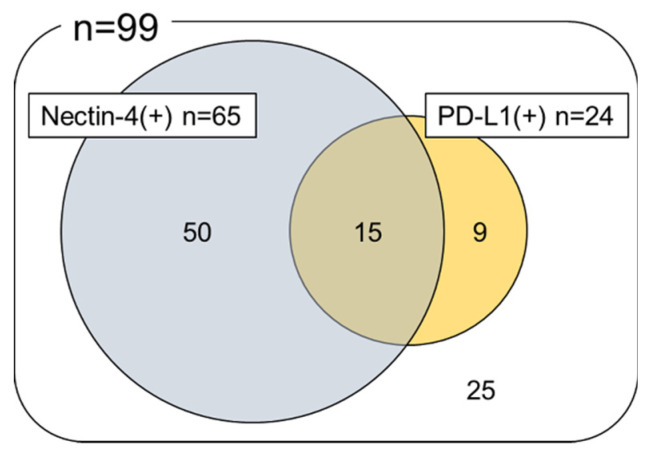
Association between the expression of Nectin-4 and Programmed Death Ligand 1 (PD-L1).

**Figure 4 ijms-21-05390-f004:**
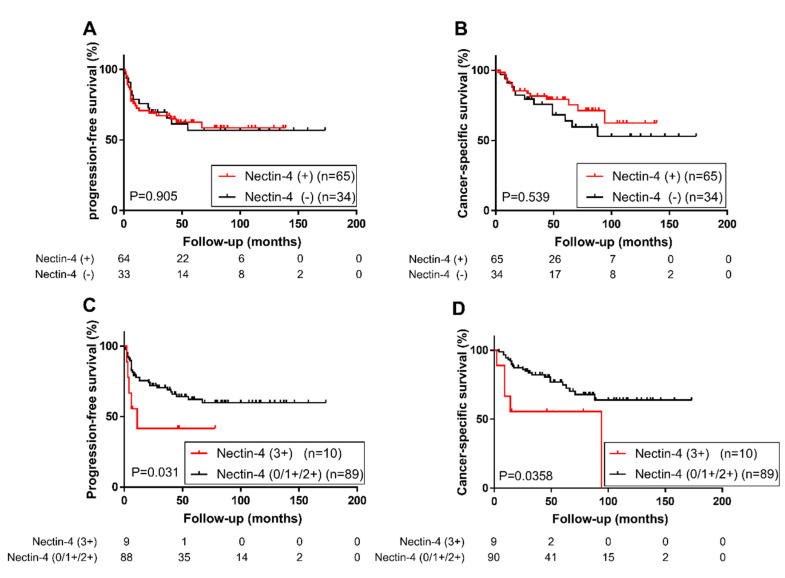
Progression-free survival (**A**) and cancer-specific survival (**B**) in 99 patients with upper tract urothelial carcinoma (UTUC) based on Nectin-4 expression (0 vs. 1+/2+/3+). Progression-free survival (**C**) and cancer-specific survival (**D**) in 99 patients with UTUC based on Nectin-4 expression (0/1+/2+ vs. 3+).

**Table 1 ijms-21-05390-t001:** Clinicopathologic characteristics and outcome of patients with upper tract urothelial carcinoma (UTUC).

Variable	
Age (year), median (range)	71 (48–87)
Sex, *n* (%)	
Male	60 (60.6)
Female	39 (39.4)
Laterality, *n* (%)	
Right	43 (43.4)
Left	56 (56.6)
Tumor location, *n* (%)	
Renal pelvis	45 (45.5)
Ureter	50 (50.5)
Both	4 (4.0)
Tumor grade, *n* (%)	
Low grade	15 (15.2)
High grade	84 (85.9)
Pathological T stage, *n* (%)	
pTa	19 (19.2)
pT1	18 (18.2)
pT2	8 (8.1)
pT3	48 (48.5)
pT4	6 (6.1)
Lymphovascular invasion, *n* (%)	
No	59 (59.6)
Yes	40 (40.4)
Lymph node metastasis, *n* (%)	
pN0	84 (84.8)
pN+	12 (12.1)
pNx	3 (3.0)
Adjuvant chemotherapy, *n* (%)	
No	63 (63.6)
Yes	26 (26.3)
Progression, *n* (%)	
No	61 (61.6)
Yes	38 (38.4)
Follow-up (month), median (range)	37 (1–173)

**Table 2 ijms-21-05390-t002:** Association of Nectin-4 expression with clinicopathological characteristics of UTUC.

Variable	Nectin-4 Expression, *n* (%)				*p*-Value	
	**0**	**1+**	2+	3+	0 vs. 1+/2+/3+	0 vs. 3+
Sex					0.134	1.000
Male	17 (28.3)	20 (33.3)	18 (30.0)	5 (8.3)		
Female	17 (43.6)	11 (28.2)	6 (15.4)	5 (12.8)		
Tumor location					1.000 ^a^	0.457
Renal pelvis	16 (35.6)	13 (28.9)	10 (22.2)	6 (13.3)		
Ureter	18 (36.0)	17 (34.0)	12 (24.0)	3 (6.0)		
Both	0 (0.0)	1 (25.0)	2 (50.0)	1 (25.0)		
Tumor grade					0.376	0.177
Low grade	7 (46.7)	6 (40)	2 (13.3)	0 (0.0)		
High grade	27 (32.1)	25 (29.8)	22 (26.2)	10 (11.9)		
Pathological T stage					0.829 ^b^	1.000
pTa	7 (36.8)	3 (15.8)	6 (31.6)	3 (15.8)		
pT1	5 (27.8)	7 (38.9)	6 (33.3)	0 (0.0)		
pT2	4 (50.0)	2 (25.0)	1 (12.5)	1 (12.5)		
pT3	15 (31.3)	17 (35.4)	11 (22.9)	5 (10.4)		
pT4	3 (50.0)	2 (33.3)	0 (0.0)	1 (16.7)		
Lymphovascular invasion					0.831	1.000
No	21 (35.6)	17 (28.8)	15 (25.4)	6 (10.2)		
Yes	13 (32.5)	14 (35.0)	9 (22.5)	4 (10.0)		
Lymph node metastasis					0.745 ^c^	0.577
pN0	29 (34.5)	26 (31.0)	21 (25)	8 (9.5)		
pN+	3 (25.0)	5 (41.7)	2 (16.7)	2 (16.7)		
pNx	2 (66.7)	0 (0.0)	1 (33.3)	0 (0.0)		

^a^ Renal pelvis vs. Ureter; ^b^ pTa+pT1 vs pT2+pT3+pT4; ^c^ pN0 vs. pN+.

**Table ijms-21-05390-t003a:** 

	Progression-Free Survival						Cancer-Specific Survival					
Variable	Univariate			Multivariate			Univariate			Multivariate		
	HR	95%CI	*p*-Value	HR	95%CI	*p*-Value	HR	95%CI	*p*-Value	HR	95%CI	*p*-Value
All cases (*n* = 99)												
Sex (male/female)	1.33	0.69–2.69	0.396				1.33	0.64–2.97	0.457			
Age (70</≤70)	1.38	0.73–2.70	0.329				1.65	0.79–3.57	0.182			
Tumor location (renal pelvis/ureter)	0.78	0.40–1.54	0.500				0.84	0.39–1.72	0.608			
Tumor grade (high/low)	4.44	1.35–27.38	0.010	4.38	1.28–27.58	0.015	7.77	1.66–138.52	0.005	5.76	1.21–103.12	0.024
pT stage (MI/NMI)	17.31	5.26–106.71	<0.001	11.35	3.23–72.07	<0.001	1.09 × 10^10^	13.52–Inf	<0.001	7.37 × 10^9^	7.70–Inf	<0.001
Lymphovascular invasion (yes/no)	5.84	2.96–12.34	<0.001	2.46	1.15–5.52	0.020	5.62	2.58–13.49	<0.001	2.20	0.93–5.23	0.073
Lymph node metastasis (yes/no)	4.40	1.95–9.00	<0.001	2.31	0.95–5.28	0.065	2.74	1.08–6.09	0.035	0.95	0.35–2.55	0.914
Nectin-4 strong expression	2.51	0.94–5.61	0.064	2.73	0.97–6.65	0.056	2.69	0.90–6.50	0.072	2.10	0.64–5.81	0.179

**Table ijms-21-05390-t003b:** 

	Overall Survival					
Variable	Univariate			Multivariate		
	HR	95%CI	*p*-Value	HR	95%CI	*p*-Value
All cases (*n* = 99)						
Sex (male/female)	1.21	0.59–2.62	0.608			
Age (70</≤70)	1.74	0.84–3.75	0.134			
Tumor location (renal pelvis/ureter)	0.89	0.43–1.84	0.757			
Tumor grade (high/low)	8.05	1.72–143.58	0.004	5.99	1.26–107.27	0.019
pT stage (MI/NMI)	1.09 × 10^10^	14.00–Inf	<0.001	7.70 × 10^10^	8.53–Inf	<0.001
Lymphovascular invasion (yes/no)	5.02	2.38–11.51	<0.001	1.97	0.87–4.70	0.105
Lymph node metastasis (yes/no)	2.63	1.04–5.81	0.041	0.95	0.35–2.55	0.914
Nectin-4 strong expression	2.60	0.88–6.25	0.081	2.09	0.71–6.14	0.181

Abbreviations: HR, hazard ratio; CI, confidence interval; MI, muscle-invasive; NMI, non–muscle-invasive; Inf, Infinity.

**Table ijms-21-05390-t004a:** 

	Progression-Free Survival						Cancer-Specific Survival					
Variable	Univariate			Multivariate			Univariate			Multivariate		
	HR	95%CI	*p*-Value	HR	95%CI	*p*-Value	HR	95%CI	*p*-Value	HR	95%CI	*p*-Value
High-risk group * (*n* = 50)												
Sex (male/female)	1.35	0.68–2.81	0.394				1.23	0.58–2.75	0.601			
Age (70</≤70)	1.11	0.57–2.23	0.771				1.57	0.75–3.48	0.238			
Tumor location (renal pelvis/ureter)	0.95	0.47–1.90	0.894				0.97	0.45–2.04	0.932			
Tumor grade (high/low)	6.96	1.49–123.94	0.008	6.30	1.34–112.50	0.014	6.41	1.36–114.68	0.014	5.73	1.20–102.73	0.025
Nectin-4 strong expression	3.79	1.37–9.11	0.013	3.32	1.20–7.98	0.027	2.78	0.92–6.87	0.067	2.31	0.76–5.77	0.128

**Table ijms-21-05390-t004b:** 

	Overall survival					
Variable	Univariate			Multivariate		
	HR	95%CI	*p*-Value	HR	95%CI	*p*-Value
High-risk group * (*n* = 50)						
Sex (male/female)	1.12	0.54–2.44	0.768			
Age (70</≤70)	1.69	0.80–3.67	0.176			
Tumor location (renal pelvis/ureter)	1.04	0.50–2.17	0.909			
Tumor grade (high/low)	6.68	1.42–119.40	0.011	6.01	1.26–107.63	0.020
Nectin-4 strong expression	2.73	0.91–6.71	0.072	2.26	0.75–5.61	0.136

Abbreviations: HR, hazard ratio; CI, confidence interval; * high-risk group is defined as patients with pathological stage higher than pT3 or presence of lymphovascular invasion or positive lymph nodes.

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
