# Peer review of "Expression of Nectin-4 and PD-L1 in Upper Tract Urothelial Carcinoma"

_ijms, 2020, doi:10.3390/ijms21155390_

Round 1
Reviewer 1 Report
Authors analysed the expression of Nectin4 and PDL1 in samples of human upper tract urothelial carcinoma (UTUC) by IHC linked to a scoring method. Nectin4 is a known tumour biomarker and it is fair to test its expression in UTUC, but there’s no evidence of its interaction with PDL1, so it is not clear to this reviewer why the authors also tested PDL1 expression instead of ErbB2 or ITGB!, known interactors of Nectin4. In my opinion the quality of the manuscript would not suffer of deleting all the PDL1 stuff and concentration in Nectin4 expression only.
On the other hand, it is difficult to ascertain the importance of Nectin4 overexpression in the UTUC progression from the data here presented since most comparisons are made of 0 vs 1+, 2+ and 3+ groups together (Table 2 and Figure 4a,b) when only significant results are obtained after comparing 3+ with 0,1+ and 2+ together (Figure 4 b,c). The authors should also present data for the 0 vs 3+ comparisons for Table 2 and Figure 4a,b
Another main concern is that the authors claim that the scoring method is the product of the staining intensity and the percentage of stained cells. Nevertheless, in Figure 1 one can only see the global staining of the field and furthermore it is difficult to distinguish among weak and moderate stainings. How this score is calculated, and especially how the percentage of stained cells is determined should be explained with more detail. Would it change the author’s claims combining the weak and moderate samples into a single group?. And what about comparing only the no-expression group with the high overexpression group? Giving the potential interest of this work I would also recommend to test by qPCR at least the control vs. higher Nectin4 groups to have more reliable quantitative data for the subsequent analysis.
Mijnor concerns.
- The claim that Nectin-4 expression was independent of PD-L1 expression is quite surprising. Should we expect that both were co-expressed?
-Figure 4 should be redrawn at a scale that facilitated its lecture, in its present presentation it is too small and difficult to read
Reviewer 2 Report
Please see attached file.

Round 2
Reviewer 1 Report
The author's have made a hard effort to improve the manuscript following my indications.